# Practical Analysis of BIM Tasks for Modular Construction Projects in South Korea

**Myungdo Lee [1], Dongmin Lee [2] , Taehoon Kim [3] and Ung-Kyun Lee [4],***

[1]   Research and Development Center, Yunwoo Technology Co., Ltd., Seoul 058054, Korea;
    md.lee@yunwoo.co.kr
[2]   Department of Civil and Environmental Engineering, University of Michigan, Ann Arbor, MI 48109, USA;
    dongminl@umich.edu
[3]   Research Institute for Mega Construction, Korea University, Seoul 02841, Korea; kimth0930@korea.ac.kr
[4]   School of Architecture, Catholic Kwandong University, Gangwon-Do 25601, Korea
*   Correspondence: uklee@cku.ac.kr; Tel.: +82-33-649-7548

**Abstract:** Building information modeling (BIM) and modular construction are important technologies for construction industry sustainability. This study proposes a relational matrix of key activities and BIM tasks of modular construction projects to analyze practical BIM tasks in Korea. To achieve this objective, 11 key activities and eight BIM tasks are identified through a comprehensive literature review and expert interviews. Then, the relational matrix of key activities and BIM tasks is proposed, and the BIM tasks in the matrix are analyzed in terms of necessity and efficiency using 5-point Likert scales. Finally, the matrix with the BIM utilization index is suggested. As a result, the average BIM utilization index is 0.80 in the off-site phase, and the index results show that 3D shop drawings have the highest index. In the on-site phase, the average BIM UI is 0.73 and the integration of a 4D model with quantity take-off is the most efficient at 0.85. Additionally, from the decision-maker's perspective, the priority through the index presented helps in making decisions and in practical BIM execution planning. The proposed matrix is a practical reference for decision-makers considering the application of BIM in modular projects, and it contributes to a sustainable construction industry.

**Keywords:** modular construction; building information modeling; off-site construction; on-site construction; 5-point Likert scales

## 1. Introduction

Modular construction and building information modeling (BIM) are recognized as important technologies to overcome the current crisis in the construction industry [1,2]. BIM and modular technologies are highly interrelated and can be applied together to maximize profits for the construction industry [2,3]. Therefore, it is recommended that BIM and a modular method be used jointly to improve the performance of the construction industry [4]. Especially in South Korea, in the "Smart Construction Technology Roadmap for Construction Productivity Innovation and Safety Reinforcement" announced by the Ministry of Land, Infrastructure and Transport in October 2018, BIM and modular construction have been suggested as key elements in technologies for improving the construction industry. In addition, because the construction industry generates more waste and consumes more energy than other industries, there is a need for measures to improve these aspects [1–4]. To this end, it is necessary to eliminate inefficiency in the construction process. Off-site construction, such as the modular method, and the use of BIM are suggested as available options [2–7].

Modular construction, which is gaining attention in the construction industry worldwide, is a typical off-site construction method [1–3,8–11]. It is a concept that can eliminate time, cost, and material

inefficiency in conventional on-site construction by performing about 60% to 80% of the total process at the factory [11]. The Korean construction industry has been conducting projects such as barracks, school buildings, and small- and medium-sized apartment buildings, led by modular construction manufacturers. Recently, because of the increasing interest by large construction companies and public clients, attempts have been made to expand this method to middle- to high-rise buildings and large multi-unit housing projects [11].

BIM is a concept that can support various tasks such as design reviews, constructability reviews, and process and construction cost simulations through the digital visualization of target buildings based on 3D modeling [12]. In particular, off-site construction methods such as modular construction can be applied to improve construction quality, reduce construction time and costs, and enhance productivity [1–11].

BIM is perceived to be one of the new technologies capable of accompanying modular construction [1,2]. BIM applications seem to confer the functionality to support and complement the modular method and reveal its potential [3]. Therefore, BIM and the modular method are two paradigms that have been suggested to address potential issues such as lower efficiency and productivity and have been increasingly applied jointly in the construction industry [2,3].

However, systematic analysis from a practical perspective on the BIM implementation items in the modular projects has not been conducted. Although several previous studies have shown that BIM has been applied to construction projects to effectively support various activities, in modular projects, reasonable detailed utilization criteria for BIM application are not defined. This means that for decision-makers considering the use of BIM, there is no decision-support model for questions regarding whether the various possibilities for using BIM tools are reasonable for modular construction projects. The various functions provided by BIM can be applied comprehensively, regardless of the type of project and construction method. However, it would be a more valuable decision from a sustainability perspective if necessary or efficient functions were allocated and presented in consideration of the project's characteristics. In particular, from a practical perspective, the application of BIM requires a cost; therefore, it is a crucial issue to select an appropriate use among various BIM tasks and to achieve an optimal effect.

Accordingly, the purpose of this study is to conduct a practical analysis of BIM tasks for key activities in each phase of a modular construction project. Practical analysis first begins with the assumption that it is unreasonable to apply BIM to activities in modular construction without a clear objective. Additionally, this study selects key activities in modular construction that are recognized as important management factors and then proposes BIM tasks by considering which of them can be effectively applied to these key activities. Finally, the relational matrix of key activities and the BIM tasks for the modular construction project are proposed. The results of this study support decision-making for rational BIM utilization in actual modular projects. Modular construction and BIM can help the construction industry fully create the goals of sustainability [12–14]. Moreover, if BIM is strategically applied to modular projects, the effect could be maximized. The matrix proposed in this study may also contribute. The BIM tasks assigned to the key activities presented in this study effectively contribute to the productivity improvement of the activities, and this improvement is expected to serve as a catalyst for sustainability by modular construction and BIM. Ultimately, this study contributes to the sustainability of the construction industry by promoting modular construction and BIM. The conceptual diagram of this research is shown in Figure 1.

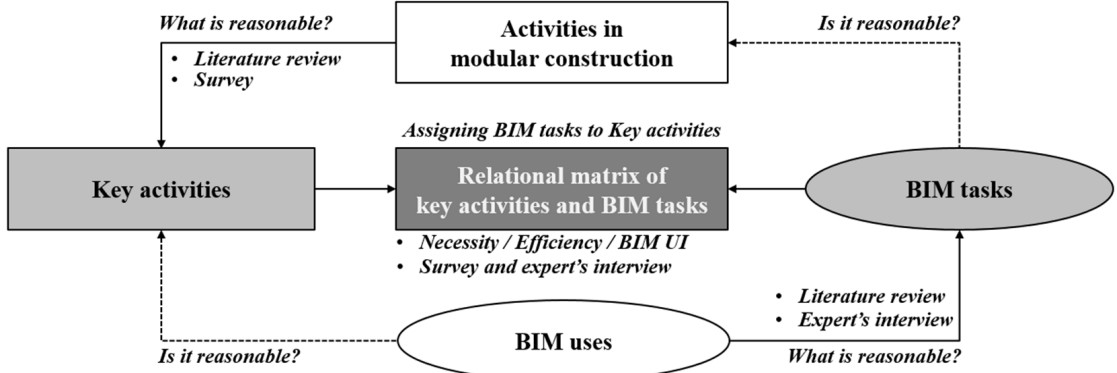

**Figure 1.** Conceptual diagram of this research.

## 2. Research Methodology

There are numerous activities in modular construction, and the application of various ways to apply BIM can be considered in this activity. However, key activities and BIM tasks need to be derived to consider a practical implementation from the perspective of reasonability. Key activities are defined as the activities of the modular project that must be managed or require improvement. BIM tasks refer to the BIM service item implemented in the actual project among the various elements of BIM, and each BIM task is an item in the BIM service contract that incurs separate costs. The BIM tasks presented in this study targeted additional BIM tasks after the completion of the basic 3D BIM modeling work. The process of deriving key activities and BIM tasks and finally presenting and reviewing the relational matrix was based on surveys and interviews with modular construction experts who understand BIM in Korea. The detailed methods and procedures of this study were as follows, and the schematics of research methodologies are shown in Figure 2.

(1)  In preliminary considerations, the BIM application is reviewed through previous studies. Then, the concept of the modular construction method and the current status of BIM in Korea are summarized.

(2)  Key activities and BIM tasks for modular projects are derived. The key activities are derived by dividing the off-site and on-site phases throughout the survey results of a 5-point Likert scale for experts as items above average. The BIM tasks are derived through literature review and interviews with experts in BIM and modular construction.

(3)  The relational matrix for key activities and BIM tasks is proposed. First, a BIM task that can be effective when applied to each key activity is selected. Second, BIM tasks assigned to key activities of the matrix are selected as BIM tasks being presented by more than half of the experts. Finally, the experts review the matrix and confirm it.

(4)  The relational matrix based on the BIM task index is proposed. The necessity and efficiency index of the BIM tasks assigned to each key activity are deduced by modular construction experts using a survey of 5-point Likert scales. Finally, the BIM utilization index (BIM UI) is presented. The practicality of the proposed matrix is evaluated and discussed through expert interviews.

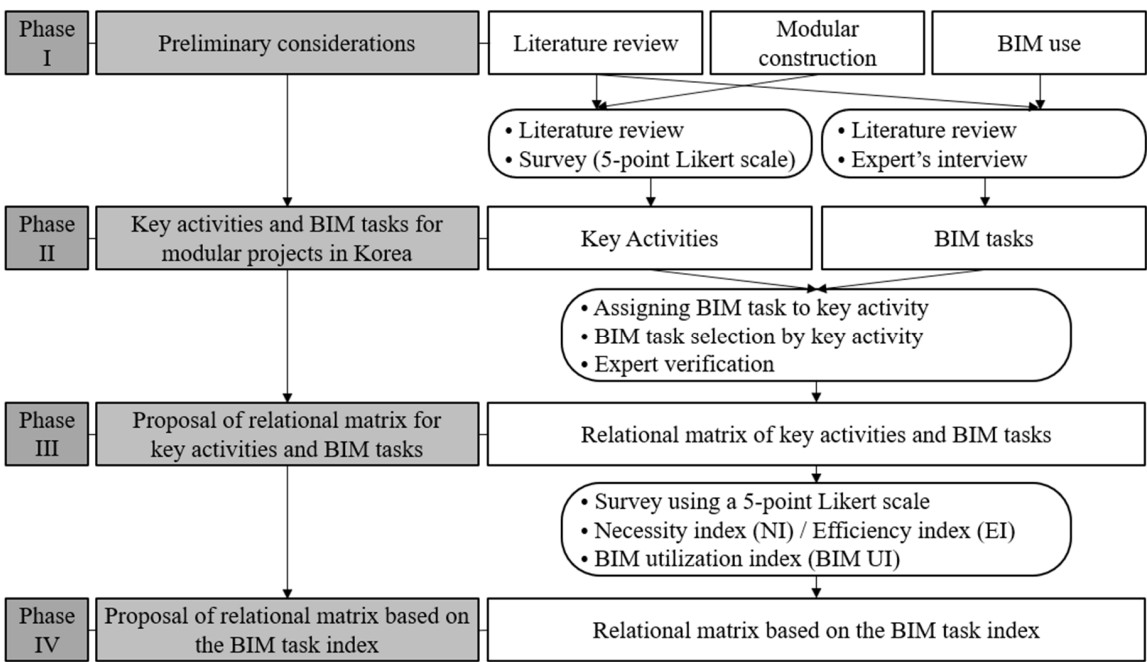

**Figure 2.** Schematic of research methodology.

## 3. Preliminary Considerations

### 3.1. Literature Review

With increasing interest in the modular construction method, studies on the application of BIM in the design, off-site fabrication, and on-site construction phases have been continuously conducted since 2010. In particular, studies on the applicability of BIM in the off-site fabrication phase are actively conducted according to the characteristics of the off-site construction method [2–4] and sustainability [5–7]. Table 1 summarizes the BIM-related studies on the modular construction method.

**Table 1.** Literature review.

| Phase | Reference | Detailed Content |
|---|---|---|
| Off-site | Alwisy and Al-Hussein (2010) [15] | Methodology of parametric modeling |
| | Moghadam et al. (2012) [16] | BIM-based schedule model for off-site manufacturing |
| | Ezcan et al. (2013) [13] | Research review of BIM and off-site manufacturing |
| | Suh and Yun (2014) [17] | Analyses of BIM-based finish material placing |
| | Abanda et al. (2017) [1] | Analyses of BIM impact for off-site manufacturing |
| | Lee (2017) [14] | Development of BIM-based 4D simulation framework |
| On-site | Han et al. (2012) [18] | Automated post-simulation visualization |
| | Han (2014) [19] | BIM-based motion planning of mobile crane |
| | Han et al. (2015) [20] | BIM-based 3D visualization of mobile crane operations |
| | Lee (2019) [11] | BIM utilization for construction planning |
| Overall | Lu and Korman (2010) [9] | Analyses of BIM impact on the design phase |
| | Korman and Lu (2011) [21] | Analyses of BIM implementation for MEP |
| | Nawari (2012) [10] | Analyses of the BIM process |
| | Lee and Lim (2012) [22] | Review of the BIM modeling process |
| | Solnosky et al. (2014) [23] | Analyses of the BIM process and tasks |
| | Samarasinghe et al. (2015) [24] | Analyses of BIM implementation on the design phase |
| | Lee and Lee (2019) [25] | BIM uptake for on/off-site construction |

Reviewing the representative studies for each phase, a study related to the performance of BIM library [22], a study on the system development for supporting BIM design [26], a study on the applicability

of BIM for quantity take-off [27], and studies on BIM application for the improvement of schedule and product quality [9,21,24] have been conducted for the design phase. For the off-site fabrication phase, studies on the process simulation for production management and process management [14,16] and a study for reducing manufacturing error [17] have been conducted. Lastly, for the on-site construction phase, studies on lifting planning through visualization of lifting equipment considering the importance of lifting operations [18–20] and studies on the review of BIM utilization for on-site construction planning [11,25] have been conducted. In addition, reviews of BIM applicability [1] and studies on the definition of the BIM application process and tasks [10,15,23] have been conducted for whole phases.

As a result of reviewing the previous studies, various BIM utilization methods and applications are proposed for each phase of modular construction. However, no study has been conducted to support decision-making with regard to which of the various BIM tasks should be applied when deciding whether BIM should be used in a real project. This is an important issue from practical perspectives, such as cost input and application efficiency of BIM. In other words, it is not possible to apply all the various BIM tasks presented in the existing studies. BIM application also needs to be considered in realistic terms if it requires the development or application of new systems other than commercial BIM software. Therefore, it is necessary to identify the parts that need improvement in the current modular project and to propose appropriate BIM tasks for them.

### 3.2. Modular Construction in Korea

Modular construction is a typical off-site construction method that can perform about 60% to 80% of the total construction work in the factory and is based on steel frame modules [11,14,25]. Modular construction is largely divided into three phases—design phase, off-site fabrication phase, and on-site construction phase [11,25]—and is expected to improve construction quality, reduce construction time, and reduce construction costs [1,10]. In South Korea, modular construction has been applied to schools, barracks, and small apartment buildings since early 2000. Recently, studies on technology development for its application to middle- to high-rise buildings have been conducted [11,14,25]. Most of the modules weigh more than 10 tons because buildings use concrete for the floor heating system in Korea. These features require high-performance lifting equipment. In addition, buildings with 13 floors and above must meet the legal criteria of having 3 h of fire resistance, resulting in a rise in construction difficulty and costs. Nevertheless, the planning of such large-scale modular projects is actively encouraged by public clients and large construction companies because modular construction is considered the future of the construction industry [11,25].

The basic processes of modular construction projects are as follows. First, the design phase is completed through collaboration between the design office and the modular manufacturer. The modular manufacturer proceeds with factory production by establishing a factory production schedule and ratio in consideration of the overall schedule and project status. At this point, the excavation and the foundation work are carried out on-site in accordance with the production schedule of the modules. Modules manufactured at the factory are transported to the site in the appropriate sequence and stacked by lifting equipment. Then, the stacked modules are connected. Work on the utilities, internal and external finishing, and other construction tasks are then carried out [25].

In Korea, a BIM application for modular projects is currently in a trial phase. The commercial BIM software converts 2D computer-assisted drawing (CAD) objects into 3D models and utilizes only basic functions such as error review, image extraction, and process plan simulation of drawings based on 3D functions provided by the software. These BIM utilizations are mainly used by design companies; few modular manufacturers and contractors have utilized BIM to date. There are several limitations to applying BIM in modular manufacturing and construction companies. First, there is a lack of BIM experts. Second, these companies are skeptical about the increase in costs resulting from BIM application. Third, modular manufacturing and construction companies also tend to preserve existing methods rather than applying new technologies such as BIM because there is a fundamental

lack of planned modular projects. Therefore, basic and clear application criteria for BIM are needed rather than a drastic change through the introduction of BIM.

### 3.3. BIM Use in Korea

BIM is used widely in the construction industry around the world and is recognized for its effectiveness. In Korea, BIM has been actively used for about 10 years; public construction projects are required to apply BIM, and private construction projects are also actively applying this technology.

BIM is used for a variety of purposes in construction projects, and its use and role have been proposed in numerous previous studies. The BIM Project Execution Planning Guide [28] has been referenced by many BIM guidelines and standards, and the 24 items that constitute BIM presented in this guide are used in various studies related to BIM application methods, criteria, guidance, and evaluation. However, from a realistic perspective, this does not mean that the items are utilized in all practical aspects or that they completely replace existing tasks. Several items indicate the usability of BIM in a theoretical sense only, and additional supporting systems may be required. There is thus a difference in definition and scope between the proposed BIM tasks and the actual needs of the project. These differences may cause confusion when considering the application of BIM to modular projects.

Additionally, Korea has so far remained at the stage of building a BIM model by utilizing BIM software, following the previous approach of working on 2D CAD objects. This is one of the main reasons for limiting the various uses of BIM to the theoretical aspects. The practical use and effects of BIM will occur when the user or organization has undergone thorough planning and preparation for BIM application. However, the application of BIM has to date been attempted without realistic plans and guides, resulting in a negative perception of its capabilities. Therefore, it is necessary to define BIM use in a way that will be effective for modular projects and to define the appropriate BIM tasks.

## 4. Key Activities and BIM Tasks for Modular Projects

### 4.1. Key Activities

Key activities of a modular project were derived for practical BIM application considering efficiency. The key activities presented in this study targeted activities that needed management or improvement among all activities related to off-site fabrication and construction of the modular project. First, we collected the fabrication and construction plans of 11 projects from three leading modular manufacturers in Korea and reviewed all activities included in the modular projects. The basic activities of modular projects are classified into off-site fabrication and on-site construction and consist of about 18 to 22 activities. Based on the results of the literature reviews [25,29], the activities in modular projects were finally classified into 22 items.

A survey was conducted to select key activities from the 22 activities presented. The subjects of the survey were 19 experts on modular projects in Korea. The average length of experience was 5.9 years. The experts were asked to grade the importance of the 22 activities on a 5-point Likert scale: "1 = Unimportant, 2 = Of little importance, 3 = Moderately important, 4 = Important, and 5 = Highly important". First, reliability verification was performed to verify the survey results. This analysis was conducted using IBM SPSS Statistics v25, and internal consistency was assessed using Cronbach's alpha coefficient. The alpha coefficient has a value between 0 and 1, and generally, if it is 0.6 or higher, it is considered reliable [30]. In this survey, the value is 0.951 for the survey, and it seems to have sufficient internal consistency. According to the survey, the average of all activities was 3.955, and 11 items above the average were selected as key activities. This does not mean that items below the average are less important. However, the mean difference between the 11th (4.08) and 12th (3.75) activities is large, and the standard deviations of the 12th and later activities are greater than 0.79, while the standard deviation for key activities is distributed under 0.66. Therefore, the results are considered to be reasonable. Table 2 summarizes the survey results.

**Table 2.** Key activities in off-site and on-site phases.

| Phase | Code | Classification | Mean | Standard Deviation |
|---|---|---|---|---|
| Off-site | C1 | Schedule management of overall off-site fabrication | 4.833 | 0.389 |
| | C2 | Planning and management of the module fabrication process | 4.083 | 0.515 |
| | C3 | Quality management of the joint assembly | 4.333 | 0.492 |
| | C4 | Schedule management of module structure fabrication | 4.417 | 0.515 |
| | C5 | Quality management of metal/door/window/siding/roof/finishing work | 4.667 | 0.492 |
| | C6 | Quality management of mechanical/electrical/plumbing work | 4.417 | 0.515 |
| On-site | C7 | Schedule management of overall on-site construction | 4.750 | 0.452 |
| | C8 | Planning and management of lifting work | 4.667 | 0.651 |
| | C9 | Quality management of module-to-module joining work | 4.250 | 0.622 |
| | C10 | Quality management of metal/door/window/siding/roof/finishing work | 4.417 | 0.669 |
| | C11 | Quality management of mechanical/electrical/plumbing work | 4.583 | 0.515 |

The key activities presented in Table 2 are the results of a 5-point Likert scale survey concerning 22 activities derived from the literature review. Among the 22 items derived in the first round through the literature review, 11 items derived above the mean are presented as the key activities. As a result of the survey, six activities were selected at the off-site fabrication phase, and five items were selected at the on-site construction phase. They are activities mainly related to schedule and quality control. The highest mean activity is the schedule of on-site construction and the schedule of off-site fabrication (C1, C7). As with other construction methods, schedule management is found to be the most important factor in modular construction. The third highest activity is the quality management of metal/door/window/siding/roof/finishing work (C5). The result was reflected in the relatively high probability of construction error because of the relatively high level of manual work. The fourth highest activity is the planning and management of lifting work (C8), reflecting the nature of the modular construction method in which modular units of 3000 mm × 6000 mm in size weighing about 10–15 ton or more on average are lifted. In particular, the lifting sequence of modules is a critical path on-site and has a significant impact on structural stability or overall scheduling. The fifth highest activity is the quality management of mechanical/electrical/plumbing work (C11), and it is considered to require more detailed work because of modularization. Other key activities are the contents of scheduling and quality control that reflect modular characteristics.

The key activities were targeted for important activities contained in construction planning documents of actual modular projects and were selected through expert surveys. The activities can be defined as having inclusiveness because if BIM is applied by focusing on the activities presented, higher application effects can be obtained.

*4.2. BIM Tasks*

In this study, the following steps were taken to define BIM tasks for modular projects. First, the BIM use items presented in the BIM Project Execution Planning Guide-version 2.2 (BIM guide) [28] and BIM utilization presented in the Basic Guide to Applying BIM for Facility Business-v2.0 (PPS guide) [31] are reviewed to perform the primary classification. At this stage, BIM tasks were selected considering the status of BIM application in Korea. For example, BIM items that are not actually performed, such as engineering analysis and cost estimation, were removed. The creation of a 3D model for site layout and conditions (T3), creation of a 4D simulation model (T4), and creation of a 4D model for the lifting plan (T6) were presented. In the second step, based on the literature review, the following items were presented: Creation of a 3D model for site layout and conditions (T3), creation of a 4D model for the lifting plan (T6), and integration of a 4D model with quantity take-off (T7). In the third step, through interviews with BIM experts in Korea, BIM tasks were suggested in consideration of the BIM service items currently being applied or attempted in actual projects in Korea, such as 3D shop drawings (T2) and integration of a 4D model with quantity take-off (T7). As a final step, interviews

were conducted with modular construction experts. Detailed 3D modeling for critical joints (T1), 3D shop drawings (T2), creation of a 4D sequence model for critical joints (T5), and integration of a 4D model with quantity take-off (T7) were presented in the interview. As a result, the BIM tasks presented at each step were finally reviewed, and the practical BIM tasks for the modular project were proposed (Table 3). The tasks are additional BIM service items for the modular project to which the basic BIM modeling is applied, and the tasks are based on 3D BIM modeling. Therefore, the tasks can be defined as a function of BIM reflecting the characteristics of the modular method.

**Table 3.** Building information modeling (BIM) tasks for modular construction project.

| Code | BIM Task | Reference | | | | |
| --- | --- | --- | --- | --- | --- | --- |
| | | BIM Guide | PPS Guide | Literature Review | BIM Experts | Modular Construction Experts |
| T1 | Detailed 3D modeling for critical joints | | | | | √ |
| T2 | 3D shop drawings | | | | √ | √ |
| T3 | Creation of a 3D model for site layout and conditions | √ | √ | √ | | |
| T4 | Creation of a 4D simulation model | √ | √ | | | |
| T5 | Creation of a 4D sequence model for critical joints | | | | | √ |
| T6 | Creation of a 4D model for the lifting plan | √ | | √ | | |
| T7 | Integration of a 4D model with quantity take-off | | | √ | √ | √ |

In Korea, most BIM services use Autodesk® Revit®, and structural modeling is mainly used with Trimble® Tekla Structure®. In this study, whole modeling was performed using Autodesk® Revit® for the integration of the entire model and the linkage of simulation.

The detailed 3D modeling for critical joints (T1) is detailed modeling of important parts in terms of structural stability or important joints to which other types of work are connected. The critical joints are selected among parts with high construction precision or where construction errors are likely to occur. If modular projects to which BIM is applied are basically established at a level of development (LOD) 300 to 350, then this part must be established at LOD 400. This task is often required in general projects because it is inefficient to build an entire project at a high LOD. The BIM task contributes to enhancing the workers' understanding and reducing construction errors. An example of the BIM task results for T1 is shown in Figure 3.

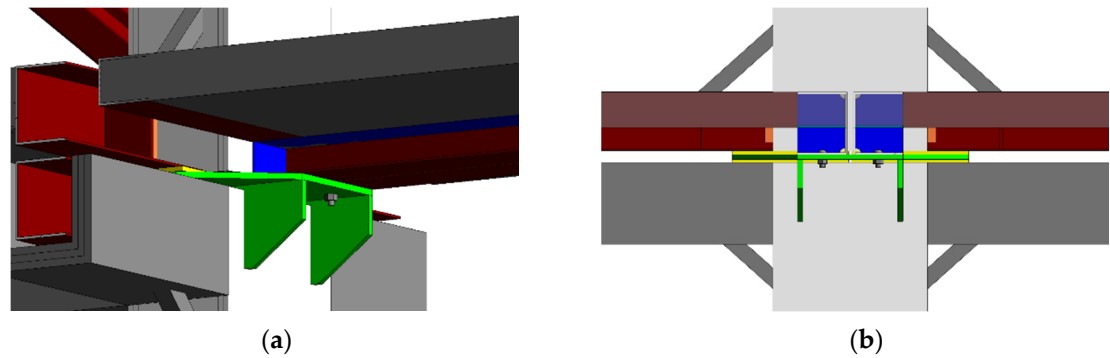

(**a**)  (**b**)

**Figure 3.** Example of detailed three-dimensional (3D) modeling for critical joints (T1): (**a**) 3D view of steel connection; (**b**) sectional plan view of steel connection.

The 3D shop drawing (T2) is a promising BIM task that has been actively requested in the field of structure and mechanical, electrical, and plumbing (MEP) BIM areas recently. This task complements the existing 2D shop drawings to enhance the understanding of workers and help improve quality and determine accurately the cost of materials needed. An example of the BIM result for T2 is shown in Figure 4.

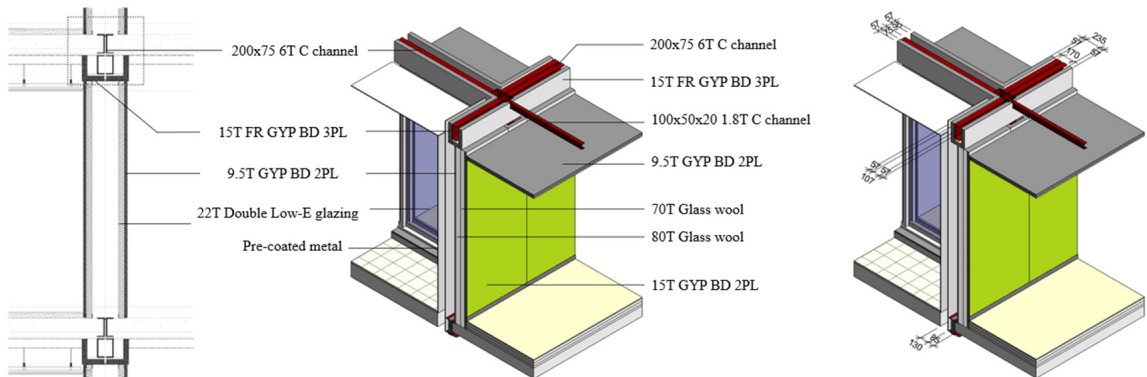

**Figure 4.** Example of T2 (part of the 3D shop BIM model).

The creation of a 3D model for site layout and conditions (T3) supports the site layout planning for the major temporary facilities such as a main entrance, field office, and modular yard. This task helps identify in advance various situations that may affect construction work, such as conditions on the surrounding roads, trees, high-voltage lines, extra space, and any obstacles. An example of the BIM task result by Autodesk® Navisworks® for T3 is shown in Figure 5.

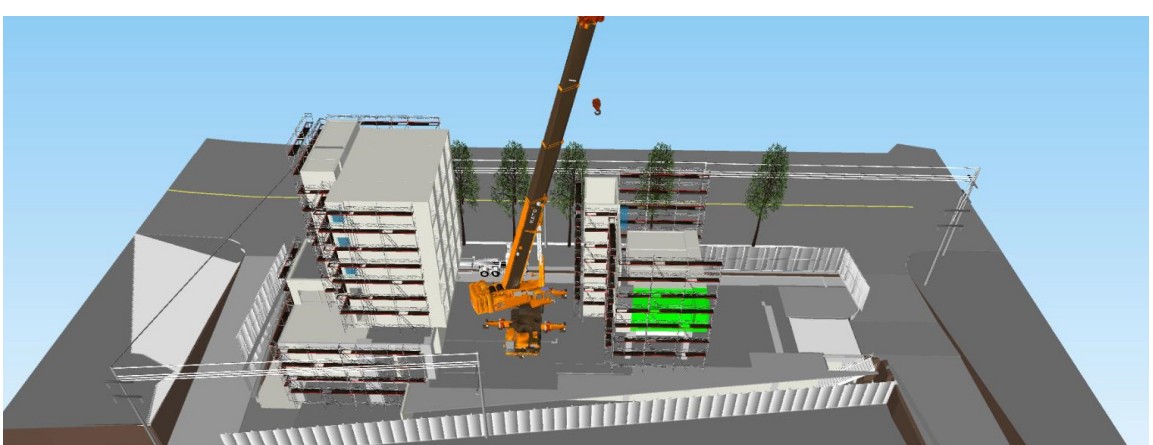

**Figure 5.** Example of T3 (part of site layout planning).

The creation of a 4D simulation model (T4) means building a 4D model that links the schedule to the 3D model of the project. This BIM task is also often required in general projects and is a powerful visualization tool that also serves as a communication tool. If the 4D simulation model for major activities is presented for the off-site and on-site phases of the modular project, it is expected to help establish and manage a schedule. The creation of a 4D sequence model for critical joints (T5) is an additional task for T1 and increases its effectiveness. The creation of a 4D model for the lifting plan (T6) reflects the characteristics of a modular method that requires a massive modular unit to be lifted. This task uses a 4D model that includes the selection, location, and movement of lifting equipment. This work helps present the most efficient lifting plans, such as the sequence of the module units and the formation of a moving line that minimizes interference with other tasks. An example of the BIM results for T6 is shown in Figure 6.

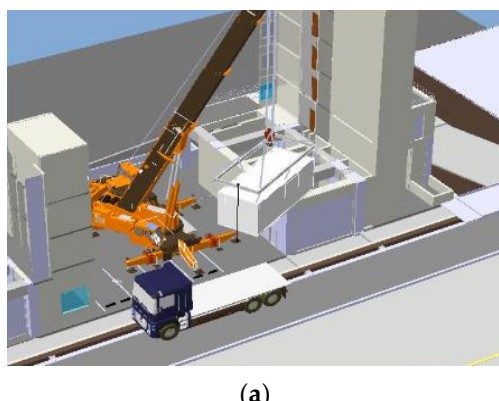　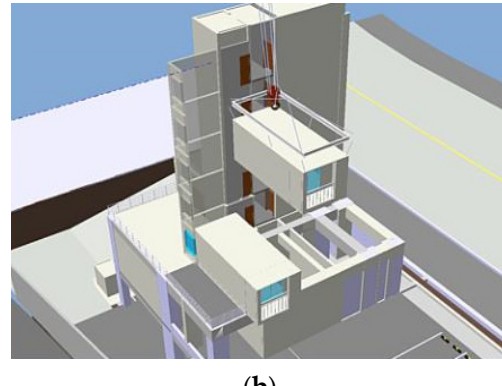

(**a**)　　　　　　　　　　　　　　　　　　　　　　　　　(**b**)

**Figure 6.** Example of T6 (part of lifting plan): (**a**) 3D view of the lifting plan; (**b**) four-dimensional (4D) sequence of the module stacking.

The last item, the integration of the 4D model with quantity take-off (T7), links the results of quantity take-off to the 4D model created for T4. It is expected that if quantity take-off items that have been injected are presented according to the progress of the work, this is of considerable assistance for material ordering, cost control, and schedule management.

### 4.3. Relational Matrix of Key Activities and BIM Tasks

If the BIM tasks are applied to the key activities, they assist in carrying out the project. However, this does not mean that the proposed BIM tasks are effective for all key activities. In addition, some of the key activities may not be major items, depending on the nature of the project or the level of participants. Therefore, the decision-maker for the project needs to select the key activities of the project and assign the appropriate BIM task to them. The relational matrix of key activities and BIM tasks is required for this decision. Based on the matrix, assigning an effective BIM task to each key activity is an important decision and increases the productivity and economics of BIM application. For this assignment, interviews were conducted with experts who derived the key activities. The interviews were conducted in three stages. First, a BIM task that can be effective when applied to each key activity was selected. Second, a relational matrix of key activities and BIM tasks was proposed by collecting the results of the experts' choices. The BIM task assigned to the key activities of this matrix was selected, with the BIM task being presented by more than half of the experts. Finally, the experts reviewed the matrix and confirmed the results were reliable. The relational matrix is shown in Table 4.

**Table 4.** Relational matrix of key activities and BIM tasks.

| Classification | | BIM Tasks | | | | | | |
|---|---|---|---|---|---|---|---|---|
| Phase | Key Activity | T1 | T2 | T3 | T4 | T5 | T6 | T7 |
| Off-site | C1 | | | | √ | | | √ |
| | C2 | √ | | | √ | √ | | √ |
| | C3 | √ | √ | | | √ | | |
| | C4 | √ | | | √ | √ | | √ |
| | C5 | √ | | | | √ | | |
| | C6 | √ | √ | | √ | | | |
| On-site | C7 | | | √ | √ | | √ | √ |
| | C8 | | | √ | √ | | √ | |
| | C9 | √ | | | | √ | √ | |
| | C10 | √ | | | | √ | √ | |
| | C11 | √ | √ | | √ | √ | | |

As a result of the interviews, 2–3 BIM tasks were assigned to each activity. In particular, T1, T4, and T5 were those most frequently assigned to each activity. This is a result that reflects the importance of the module connections because of the characteristics of the modular method. For T3, tasks were allocated to C7 and C8 as BIM tasks used in the on-site phase.

## 5. Relational Matrix Based on the BIM Task Index

### 5.1. Survey Overview

The relational metrics help the decision-maker decide which key activity of the project should be selected and which BIM task should be applied. In addition, if the numerical values of the necessity and efficiency of the allocated BIM task are presented, more practical decisions can be made. In this study, the numerical values of the necessity and efficiency for each BIM activity were presented as the BIM utilization index (BIM UI).

In this study, BIM tasks that can be usefully applied to the key activities were presented as a relational matrix. The goal is to maximize the application effect by applying the BIM tasks assigned to the key activities. From the decision-maker's perspective, information is needed on how much the BIM task assigned to the key activities is needed and to what extent the effects can be expected. In this study, the necessity and efficiency of BIM tasks were presented as indexes. Since each index was derived from a survey based on a 5-point Likert scale, the higher the index, the more it can be defined as useful. Therefore, the arithmetic mean of each index was presented as BIM UI. Therefore, the BIM UI was presented as the average value of the calculated necessity index (NI) and the efficiency index (EI), as shown in Equation (1):

$$\text{BIM utilization index (BIM UI)} = (\text{NI} + \text{EI})/2 \qquad (1)$$

To derive the BIM utilization index, a survey was conducted among domestic modular construction experts. The survey was conducted on 5-point Likert scales for the necessity and efficiency of the BIM task assigned to each key activity. First, the questionnaire asked, "What level of necessity should be given to those BIM tasks assigned to the key activity?", and respondents assessed this using a Likert scale. The second question was, "How efficient are the allocated BIM tasks compared with the existing method (non-BIM)?", and this was also assessed using the Likert scale. The results of the survey are presented with the necessity index (NI) and the efficiency index (EI) [11,25], as shown in Equations (2) and (3):

$$\text{Necessity Index (NI)} = \left( \sum_i^5 w_i \times \frac{f_i}{n} \times 100 \right) / (a \times 100), \qquad (2)$$

$$\text{Efficiency Index (EI)} = \left( \sum_j^5 w_j \times \frac{f_j}{n} \times 100 \right) / (a \times 100), \qquad (3)$$

where $i$ and $j$ represent 5-point scales from 1 to 5; $w_i$ and $w_j$ are the scale scores of necessity and efficiency given to the BIM task; $f_i$ and $f_j$ are the frequencies of the scale scores given by all experts; $n$ is the number of experts; and $a$ is the highest score of 5.

The survey was conducted on modular construction experts with basic knowledge of using BIM. The questionnaire contained a detailed description of the main activities and visual materials for the BIM tasks to enhance understanding, and the surveys were conducted face-to-face to ensure reliability. A total of 22 completed questionnaires have been collected, and the respondents have an average of 5.2 years of experience in conducting and researching modular construction projects. The 22 completed questionnaires comprise a relatively small sample size. Since Korea is in the pre-activation stage of modular construction, there was a limit to the supply and demand of experts who are skilled in modular construction and have a high understanding of BIM. Therefore, even though there was

a small number of respondents, the survey focuses on improving reliability and is judged to have sufficient persuasiveness.

*5.2. Results*

First, the reliability analysis of the survey results was performed using IBM SPSS Statistics v25, and internal consistency was assessed using Cronbach's alpha coefficient. As a result, the values were high at 0.812 for necessity and 0.941 for efficiency, indicating sufficient internal consistency.

Based on the survey results, NI, EI, and BIM UI were derived (rounded to the second decimal point) and reflected in the relational matrix (Table 4), as shown in Table 5. In addition, the average of the index for each BIM task was inserted by dividing it into off-site, on-site, and overall phases. The BIM tasks assigned to each key activity were the result of the experts' interviews, so the overall index was high. The overall index formed had a high value (avg. 0.74) because the BIM tasks assigned to each key activity were the result of the experts' interviews. For the off-site phase, T2 had the highest UI value (0.92), while the average for UI was 0.79. Comparing NI and EI, the gap was not great (0–0.04), but NI was the same (T2) or higher in all tasks. For the field phase, T7 acquired the highest UI (0.85). The gap between NI and EI was small at the off-site phase (0–0.05), and the UI average for the on-site phase was 0.72. The BIM task with the highest NI in the off-site phase was T7 for C1, T7 for C2, and T2 for C3. For EI, T2 for C3 was 0.97, which was the highest value in the entire index. In the on-site phase, T1 for C9 was the BIM task with the highest NI and EI with 0.93. The BIM tasks with the highest BIM UI in the whole phase were T2 and T7, shown to be 0.87.

**Table 5.** Relational matrix based on the BIM task index.

| Phase | Key Activity | T1 NI | T1 EI | T2 NI | T2 EI | T3 NI | T3 EI | T4 NI | T4 EI | T5 NI | T5 EI | T6 NI | T6 EI | T7 NI | T7 EI | Total NI | Total EI |
|---|---|---|---|---|---|---|---|---|---|---|---|---|---|---|---|---|---|
| Off-site | C1 | | | | | | | 0.80 | 0.77 | | | | | 0.93 | 0.90 | 0.87 | 0.84 |
| | | | | | | | | UI 0.78 | | | | | | UI 0.92 | | UI 0.85 | |
| | C2 | 0.80 | 0.80 | | | | | 0.73 | 0.77 | 0.73 | 0.80 | | | 0.93 | 0.90 | 0.80 | 0.82 |
| | | UI 0.80 | | | | | | UI 0.75 | | UI 0.77 | | | | UI 0.92 | | UI 0.81 | |
| | C3 | 0.80 | 0.83 | 0.93 | 0.97 | | | | | 0.70 | 0.67 | | | | | 0.81 | 0.82 |
| | | UI 0.82 | | UI 0.95 | | | | | | UI 0.68 | | | | | | UI 0.82 | |
| | C4 | 0.63 | 0.60 | | | | | 0.77 | 0.70 | 0.67 | 0.60 | | | 0.87 | 0.80 | 0.74 | 0.68 |
| | | UI 0.62 | | | | | | UI 0.73 | | UI 0.63 | | | | UI 0.83 | | UI 0.71 | |
| | C5 | 0.83 | 0.77 | | | | | | | 0.60 | 0.53 | | | | | 0.72 | 0.65 |
| | | UI 0.80 | | | | | | | | UI 0.57 | | | | | | UI 0.68 | |
| | C6 | 0.83 | 0.80 | 0.90 | 0.87 | | | 0.67 | 0.67 | | | | | | | 0.80 | 0.78 |
| | | UI 0.82 | | UI 0.88 | | | | UI 0.67 | | | | | | | | UI 0.79 | |
| | Total | 0.78 | 0.76 | 0.92 | 0.92 | | | 0.74 | 0.73 | 0.68 | 0.65 | | | 0.91 | 0.87 | 0.81 | 0.79 |
| | | UI 0.77 | | UI 0.92 | | | | UI 0.73 | | UI 0.66 | | | | UI 0.89 | | UI 0.80 | |
| On-site | C7 | | | | | 0.60 | 0.60 | 0.83 | 0.80 | | | 0.77 | 0.77 | 0.87 | 0.83 | 0.77 | 0.75 |
| | | | | | | UI 0.60 | | UI 0.82 | | | | UI 0.77 | | UI 0.85 | | UI 0.76 | |
| | C8 | | | | | 0.67 | 0.63 | 0.60 | 0.63 | | | 0.90 | 0.90 | | | 0.72 | 0.72 |
| | | | | | | UI 0.65 | | UI 0.62 | | | | UI 0.90 | | | | UI 0.72 | |
| | C9 | 0.93 | 0.93 | | | | | | | 0.63 | 0.63 | 0.63 | 0.63 | | | 0.73 | 0.73 |
| | | UI 0.93 | | | | | | | | UI 0.63 | | UI 0.63 | | | | UI 0.73 | |
| | C10 | 0.73 | 0.60 | | | | | | | 0.60 | 0.53 | 0.63 | 0.63 | | | 0.65 | 0.59 |
| | | UI 0.67 | | | | | | | | UI 0.57 | | UI 0.63 | | | | UI 0.62 | |
| | C11 | 0.77 | 0.73 | 0.83 | 0.80 | | | 0.57 | 0.53 | 0.60 | 0.63 | | | | | 0.69 | 0.67 |
| | | UI 0.75 | | UI 0.82 | | | | UI 0.55 | | UI 0.62 | | | | | | UI 0.68 | |
| | Total | 0.81 | 0.76 | 0.83 | 0.80 | 0.63 | 0.62 | 0.67 | 0.66 | 0.61 | 0.60 | 0.73 | 0.73 | 0.87 | 0.83 | 0.74 | 0.71 |
| | | UI 0.78 | | UI 0.82 | | UI 0.63 | | UI 0.66 | | UI 0.61 | | UI 0.73 | | UI 0.85 | | UI 0.73 | |
| Grand total | | 0.80 | 0.76 | 0.88 | 0.86 | 0.63 | 0.62 | 0.70 | 0.69 | 0.64 | 0.63 | 0.73 | 0.73 | 0.89 | 0.85 | 0.75 | 0.73 |
| | | UI 0.78 | | UI 0.87 | | UI 0.63 | | UI 0.70 | | UI 0.63 | | UI 0.73 | | UI 0.87 | | UI 0.74 | |

The average BIM UI of the off-site phase is 0.80, and for each activity, it can be interpreted as follows. The C1 was assigned T1 and T4, which have the highest BIM UI of 0.85 on average at the off-site phase. This means that the most useful results can be expected when T4 and T7 are applied to

C1. From the decision-maker's perspective, for projects where C1 is considered important, the priority through the index presented helps make decisions. The C2 was assigned T1, T4, T5, and T7. The index results showed that T7 had the highest index in C2, as with case C1. The index of T7 also had the highest value of 0.83 in C4 as a result of T7 being the most useful task for all activities of C1, C2, and C4. Since C1, C2, and C4 are activities related to schedule management, T7 was considered highly utilized. Therefore, if time control is an important consideration, the decision-maker needs to consider the application of T7. In the case of C3, the index for T2 was the highest at 0.95. T2 is the task assigned to C3 and C6 and is the task of the highest index in each activity in the off-site phase. That is, for projects where quality control is important, it is necessary to consider T2. In C5, T1 was 0.80 and T5 was 0.57. The gap between these two indexes is greater than the deviation between each task of the other activities and is the lowest average of 0.68 at the off-site phase. C5 is also an item related to quality control. T1 has a sufficient effect for C5, and T5 is considered an additional support option.

For the on-site phase, the average of BIM UI is 0.73, and the index is generally lower than that for the off-site phase. In the case of C7, T3, T4, T6, and T7 were allocated, and the average BIM UI was 0.76, the highest in the off-site phase. T7 was the highest index at 0.85 in C1, as with cases C1, C2, and C4 at the off-site phase. In the case of C8, T6 was 0.90 and the index was significantly higher than with the allocated T3 (0.65) and T4 (0.62). C8 and T6 items are related to lifting work, and T6 is the task assigned to all activities except C11 in the on-site phase, and utilization is actively considered. T1, T5, and T6 were allocated in C9, and a much higher index was derived with T1 (0.93) than with the others. C9 is an important task related to the safety of a construction site, and the application of T1 is actively considered. The C10 was assigned T1, T5, and T6; the index average was 0.62, and the lowest index from on-site was derived. C10 is the lowest priority among all key activities, and it is judged that the priority is reflected in the results of BIM UI. Finally, in the case of C11, T1 was 0.75, T2 was 0.82, T4 was 0.55, and T5 was 0.62. For C11, the application of T2 may be considered first.

## 6. Discussion

The relational matrix based on the BIM task index presented in this study is a practical reference for decision-makers considering the application of BIM to modular projects. The decision-maker can select the key activity of a modular project by characterizing the project. Subsequently, the BIM task and its index assigned to the selected activity can be reviewed, and then the decision-maker can select a BIM task.

To evaluate the practicality of the proposed matrix, five experts with more than 10 years of experience in directly carrying out modular projects were interviewed. All experts said that the proposed matrix is a practical reference for decision-making. The results of the interviews are summarized as follows. First, the key activities and BIM tasks of the matrix were presented. Second, NI, EI, and BIM UI can help make decisions. Third, selecting BIM tasks is useful for writing a clear contract because the cost of applying BIM is broken down by each task included. Finally, the application target and scope of the BIM tasks are made clear and are effective for identifying their effects. However, the limitations of the matrix must be considered. It is difficult to interpret the meaning of the difference in the index value of each BIM project. The matrix would be more useful if the effects of applying each BIM task were linked to cost, time, quality, and other important factors. Therefore, if the values of numerical gaps for each index were provided through additional statistical analysis considering those factors, the matrix would be more useful to decision-makers.

In addition, the factors and the corresponding index of the relational matrix in this study may depend on the characteristics of modular projects, decisions of the BIM manager, and job description of the project. Namely, the matrix must be flexible and extensible. Therefore, plans are in place to provide an advanced matrix by reflecting a variety of situations through continuous case studies in Korea.

## 7. Conclusions

BIM and modular methods are recognized as important technologies for sustainability [2,3]. Although they are promising, practical application guidelines are needed for them to achieve their potential. In this study, a matrix to identify optimum BIM tasks is presented for key activities in the off-site and on-site phases of modular building projects.

The relational matrix has the form of assigning seven BIM tasks to 11 key activities of a modular construction project, and the index was inserted through the results of a questionnaire survey. In a modular project that considers BIM application, the decision-maker will be able to review the key activities of that project and determine the purpose and scope of BIM application through this matrix and index.

As a future study, following the experts' opinions regarding the limitations of the matrix, we are planning to analyze the extent to which the index is valuable and develop a matrix that can present the effect of applying BIM tasks. In addition, the factors and the corresponding index of the relational matrix in this study may depend on the characteristics of modular projects, decisions of the BIM manager, and job description of the project. Therefore, the study plans to supplement the matrix by reflecting a variety of situations through continuous case studies. In particular, further work will include statistical analysis of the differences in index values of BIM tasks assigned to key activities or the differences in index values of the same BIM task assigned to each key activity.

The limitation of this study is that the survey was conducted with a small number of experts. Taking this approach assumed that the results of the survey could be trusted because modular construction experts with an understanding of BIM were selected. Further expert opinions and cases will be obtained and reflected in future studies. In addition, because analysis of all activities and tasks of the modular construction method is limited in terms of the efficiency of the research and the utilization of results, the key activities and BIM tasks were proposed. However, this does not mean that excluded activities and tasks do not need to be considered. Therefore, new items will be added, and a more comprehensive analysis will be performed through further research.

The index presented in the matrix could be used as an effective guide for a decision-maker to answer the question, "Which BIM task should be applied to which activity?" Since the proposed BIM tasks have individual costs, the matrix also helps secure economic feasibility for the BIM application. If each BIM task applied helps manage a key activity, it helps to successfully carry out the modular project. Ultimately, this study will contribute to enabling the realistic application of sustainable technologies such as BIM and modular construction, and thus to sustainability.

**Author Contributions:** Conceived and designed the research, performed data analysis, and wrote the manuscript, M.L. and T.K.; contributed to the research procedure, developed a detailed research concept, and revised the manuscript, U.-K.L. and D.L. All authors have read and agreed to the published version of the manuscript.

**Funding:** This research received no external funding.

**Acknowledgments:** This research was supported by a grant (20RERP–B082884-07) from the Residential Environment Research Program funded by the Ministry of Land, Infrastructure and Transport of the Korean government.

**Conflicts of Interest:** The authors declare no conflict of interest.

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
