# Peer review of "Practical Analysis of BIM Tasks for Modular Construction Projects in South Korea"

_sustainability, doi:10.3390/su12176900_

Round 1

Reviewer 1 Report

  • Sample size is small. What statistical method is used for sampling? How is it determined that 22 is enough?
  • Using mean and especially standard deviation is not correct with ordinal data.
  • Inclusiveness and exclusiveness of tasks and activities could be elaborated.
  • The logic of BIM UI should be explained.
  • Possibly some non-parametric tests could be used to highlight the important results (e.g. in Table 5)

Author Response

We are very grateful for your opinions. I have faithfully reflected your opinions in the manuscript. Please see the attachment.

Reviewer 2 Report

  • the relation the authors want to start between BIM and the modular construction is weak. cause BIM is a methodology that helps improve the different phases of any project. so the tools exist and any tools could be applied without being assigned to a specific construction method such as the modular method
  • the list could be expanded or even shrank according to top the BIM manger and the job description 
  • I would rather prefer as a BIM user to know what template /software more involved in this methodology 

Author Response

(The authors gave the same response as above.)

Reviewer 3 Report

The paper is well-structured and written.

My minor revisions are:

- line 196. The references Lee and Sheeare are cited in the text in a wrong way according to journal template.

- Literature review section should be placed as main heading before the others sub-sections

- Result and Discussion should be listed in the manuscript as separate heading sections

Author Response

(The authors gave the same response as above.)

Round 2

Reviewer 1 Report

I believe further revision, especially in the methodology section, can improve the paper.

Author Response

 We have supplemented the methodology according to your opinion. First, for a clear description of the methodology, " 2. Research methodology" was newly distinguished, and supplementary explanations for the procedure and Figure 2 were added. Please see the attachment.
